# Physicochemical Characterization of Kynurenine Pathway Metabolites

**DOI:** 10.3390/antiox14050589

**Published:** 2025-05-14

**Authors:** Luca Buzásy, Károly Mazák, Balázs Balogh, Balázs Simon, Anna Vincze, György Tibor Balogh, Tamás Pálla, Arash Mirzahosseini

**Affiliations:** 1Department of Pharmaceutical Chemistry, Semmelweis University, Hőgyes Endre utca 9, 1092 Budapest, Hungary; buzasy.luca@stud.semmelweis.hu (L.B.); mazak.karoly@semmelweis.hu (K.M.); simon.balazs@semmelweis.hu (B.S.); vincze.anna@semmelweis.hu (A.V.); balogh.gyorgy.tibor@semmelweis.hu (G.T.B.); 2Center for Pharmacology and Drug Research & Development, Semmelweis University, 1085 Budapest, Hungary; balogh.balazs@semmelweis.hu; 3Department of Organic Chemistry, Semmelweis University, 1092 Budapest, Hungary

**Keywords:** kynurenine, metabolism, p*K*_a_

## Abstract

The kynurenine pathway is a significant metabolic route involved in the catabolism of tryptophan, producing various bioactive metabolites with crucial roles as antioxidants in immune regulation and neurobiology. This study investigates the acid-base properties of picolinic acid, kynurenic acid, kynurenine, and 3-hydroxykynurenine, utilizing computational simulations and experimental techniques, including potentiometric and nuclear magnetic resonance titrations. The results reveal distinct p*K*_a_ values, with kynurenic acid exhibiting a single dissociation step around 2.4, while kynurenine displays three dissociation steps governed by interactions between its functional groups. Additionally, 3-hydroxykynurenine shows overlapping dissociations in two separate pH regions, suggesting nuanced behavior influenced by its molecular structure. The analysis of intramolecular hydrogen bonding in protonation microspecies across varying pH highlights the relevance of the charge state and hydrogen transfer potential of these metabolites in the context of their radical scavenging ability. At physiological pH, most kynurenine and 3-hydroxykynurenine entities exist in zwitterionic form, with hydrogen bonding stabilizing the aromatic amino group, which may significantly influence their interactions with proteins and reactive oxygen species. This study provides critical insights into the acid-base equilibria of kynurenine pathway metabolites.

## 1. Introduction

The kynurenine pathway is the principal route for the catabolism of tryptophan [1], playing a crucial role in immune regulation, neuroprotection, and metabolic homeostasis [2]. This pathway leads to the production of several bioactive metabolites, including kynurenine and kynurenic acid, which have significant implications in neurological and inflammatory disorders [3]. Kynurenine serves as a central intermediate, undergoing enzymatic transformations [4] that influence neurotransmission, oxidative stress, and immune responses. Kynurenic acid, a key metabolite, acts as an antagonist of excitatory neurotransmitter receptors, including N-methyl-D-aspartate (NMDA) receptors, and has been implicated in neurodegenerative diseases, psychiatric disorders, and metabolic dysfunctions [5]. Understanding the regulation of this pathway and the interplay between its metabolites is critical for developing therapeutic strategies targeting kynurenine metabolism in various pathological conditions. Given its neuroprotective functions, kynurenic acid has emerged as a potential therapeutic agent for neurological and psychiatric disorders [6]. Efforts to modulate kynurenic acid levels pharmacologically—either through synthetic kynurenic acid analogs or inhibitors of kynurenine aminotransferase enzymes—are being explored as novel treatment strategies.

Kynurenine and 3-hydroxykynurenine are key intermediates in the kynurenine pathway, each playing distinct roles in immune regulation, neurobiology, and oxidative balance. Kynurenine can cross the blood-brain barrier and influence neurotransmission by modulating NMDA receptors through its conversion into kynurenic acid. Additionally, kynurenine is implicated in mood disorders, with elevated levels being associated with depression and cognitive decline [7], likely due to its role in neuroinflammation and excitotoxicity [8].

3-Hydroxykynurenine is a downstream metabolite of kynurenine, primarily generated by the enzyme kynurenine 3-monooxygenase. Unlike kynurenine, which has immunomodulatory and neuroactive properties, 3-hydroxykynurenine is mainly recognized for its pro-oxidant effects. It readily undergoes redox cycling, generating reactive oxygen species (ROS) that contribute to oxidative stress [9]. While controlled ROS production is essential for cellular signaling, excessive accumulation of 3-hydroxykynurenine has been linked to neuronal damage, mitochondrial dysfunction, and apoptosis [10].

Kynurenine and 3-hydroxykynurenine play a crucial role in ocular physiology, particularly in the lens and retina, where they contribute to the absorption of ultraviolet (UV) radiation [11]. These metabolites act as natural UV filters, protecting the eye from phototoxic damage by absorbing harmful high-energy light [12]. However, their photosensitive properties also make them susceptible to photochemical reactions, which can generate ROS and lead to oxidative stress [13]. Excessive accumulation of kynurenine-derived photoproducts in the retina has been linked to age-related macular degeneration (AMD) due to their potential to induce photoreceptor damage and inflammatory responses [14]. Understanding the balance between the protective and damaging effects of these metabolites is essential for developing therapeutic strategies to mitigate light-induced ocular disorders.

The physicochemical properties of these valuable compounds have been intensively studied, including quantitative determination [15,16,17], lipophilicity and membrane penetrating ability in various formulations [18,19], photochemistry [20] and spectroscopy [21]. However, no reliable data exists on their acid-base properties. Only the radical form of kynurenic acid [22] has been investigated to reveal a p*K*_a_ of 5.5 [23]. In this work, we seek to provide a thorough elucidation of the acid-base equilibria and spectroscopic behavior of these compounds in their native state (i.e., not in a radical or oxidized form). Other metabolites of this pathway have been thoroughly investigated, e.g., anthranilic acid [24]; however, there have been conflicting data on the p*K*_a_ values of picolinic acid [25,26] (1.0, 5.4 and 1.07, 5.25 values reported in the two citations, respectively, however no mention of experimental methods is made). Therefore, this intermediate metabolite was included in the study as well.

## 2. Materials and Methods

### 2.1. Chemicals

All chemicals were purchased from Sigma-Aldrich (Merck KGaA, Darmstadt, Germany) and were used without further purification. Deionized water was prepared using a Milli-Q Direct 8 Millipore system.

### 2.2. ^1^H NMR (Nuclear Magnetic Resonance) Spectroscopy Measurements

NMR spectra were recorded on a 400 MHz Varian Mercury Plus spectrometer equipped with a 5 mm Varian 400 Automation Triple Resonance Broadband Pulsed Field Gradient probe at 298.15 ± 0.1 K. The solvent was H_2_O/D_2_O 95:5 (*v*/*v*), the solution contained around 5 mmol/L titrand, and the ionic strength was adjusted to 0.15 mol/L. The pH values were adjusted with 2–5 mol/L HCl or NaOH (in order to avoid substantial dilution of the solution) and determined in situ by internal indicator molecules optimized for ^1^H NMR [27,28]. The sample volume was 500 μL containing DSS (3-(trimethylsilyl)propane-1-sulfonate) as a chemical shift reference. The H_2_O ^1^H signal was suppressed with a WET sequence; the average acquisition parameters for ^1^H measurements are number of transients = 8, number of points = 65,536, acquisition time = 3.33 s, and relaxation delay = 1.5 s. All spectra were processed using the MestReNova v9.0.1-13254 (Mestralab Research, S.L., Santiago de Compostela, Spain) software. NMR-pH titrations were performed once, and resulting p*K*_a_ values are reported with ± standard error of the model fit.

### 2.3. pH-Potentiometric Titrations

An ECO automatic titrator (Metrohm AG, Herisau, Switzerland) with a Metrohm 6.0234.110 combined pH glass electrode was used for the potentiometric titrations under automatic PC control. The electrode was calibrated with aqueous standard buffer solutions. Constant temperature (298.15 ± 0.1 K) was provided by a thermostated double-walled glass cell. Difference titrations were carried out in the absence (blank) and presence of the titrand. First, 2 mL of 0.1 mol/L HCl solutions were titrated with 0.1 mol/L KOH. A constant ionic strength of 0.15 mol/L was provided by the presence of KCl. Next, a titrand was added to the same volume of HCl solution and subsequently titrated with KOH. The initial concentration of the titrand was around between 3 and 6 mmol/L in the titrations. Non-linear parameter fitting provided the dissociation constants from the interpolated volume differences. Potentiometric titrations were performed in triplicates, and the resulting p*K*_a_ values are reported with ± standard error of the three fitted values.

### 2.4. UV-pH Titrations

UV spectroscopic measurements were carried out on a Jasco V-550 diode-array spectrometer (Jasco Ltd., Tokyo, Japan) with 10 mm quartz cuvettes at 298 K. The titrand was diluted with an acidic (HCl) and a basic (NaOH) stock solution to the same analyte concentration (adjusting ionic strength to 0.15 mol/L). In the case of kynurenic acid, a saturated acidic solution was diluted 10 times with HCl and NaOH, respectively, to obtain the stock solutions. Sample solutions with varying pH were prepared by mixing the two stock solutions in varying ratios. pH values were read on a Metrohm 2.780.0010 precision pH meter with a 6.0258.600 Unitrode glass Pt 1000 electrode (Metrohm AG, Herisau, Switzerland), and the pH-potentiometric system was calibrated using standard buffer solutions.

### 2.5. Statistical Analysis

The non-linear mixed-effects model regression fits for the titration data were performed using R version 4.0.5 (R Foundation for Statistical Computing, Vienna, Austria) [29] and the “nlme” package [30] using the function formulae described in the Results section. Data visualization was performed using R Studio integrated development environment version 2024.12.1 (Posit, Boston, MA, USA).

### 2.6. Molecular Dynamic and Quantum Chemistry Computations

All computations were carried out with the Schrödinger 2024-1 program package (Maestro, Schrödinger, LLC, New York, NY, USA). Structures were drawn using Schrödinger’s Maestro graphical user interface (GUI). Initial minimizations and conformational searches were completed with MacroModel [31] using OPLS4 forcefield [32], with water solvent effects simulated by the analytical Generalized-Born/Surface-Area (GB/SA) model. The Mixed torsional/Large-scale low-mode sampling with standard settings was used for conformer generation. Quantum chemistry optimizations were carried out for the selected conformers using Schrödinger’s Jaguar program with Density Functional Theory (DFT) B3LYP-D3 method with 6-31G** basis set [33], default settings were used except for the accuracy level (ultrafine instead of quick). Polarizable continuum model (PCM) water was applied as a solvation method [34]. The NMR shielding calculations were generated within Jaguar’s property tab [35].

## 3. Results

The p*K*_a_ values of kynurenine pathway metabolites were simulated using the computational tools Marvin (https://chemaxon.com/marvin, accessed on 31 January 2025) and Percepta (ACD Inc., Toronto, ON, Canada) to aid the experimental design process. The results of these simulations are illustrated in Figure 1. Subsequently, the compounds were titrated using potentiometry and ^1^H NMR titration methods. The macroscopic and microscopic acid-base parameters of the investigated compounds are sequentially presented and determined by methods described previously [36,37,38].

Picolinic acid exhibited a single p*K*_a_ value near 5.2, with both titration techniques yielding consistent results (Figure 2). The solubility of picolinic acid posed no challenge during the experiments. Although a slight systematic shift in the NMR signals was observed below pH 1.5, this indicates a dissociation step occurring below pH 1, outside the range of determination. The extreme acidic NMR titration was not attempted due to ionic strength constraints and the low relevance of such an extreme p*K*_a_ value. Extrapolation of the potentiometric titration data can be performed, however (taking advantage of the fact that every dissociation step must be of the same height), providing a rough estimate of the extreme acidic p*K*_a_ value at 0.63. However, it should be noted that the uncertainty of this value is surely greater than the standard error of the regression fit. This value was determined independently of the simulated values in Figure 1 and provides a more accurate representation of the picolinic acid p*K*_a1_ value.

The potentiometric titration of kynurenic acid was next attempted, and while solubility was no issue above pH 3, below this pH, the precipitation of the titrand hindered the measurement. The attempt at NMR-pH titration was also unsuccessful for the same reason; however, these measurements confirmed that there was no dissociation step above pH 7 until 13. Therefore, to determine the p*K*_a_ of kynurenic acid accurately, UV-pH titration was performed using the 10-fold diluted saturated solution of kynurenic acid in 0.15 mol/L HCl. The UV spectra showed an absorption maximum near 332 nm and an isosbestic point, indicating only one dissociation step. In order to gather accurate data, the UV-pH titration was repeated with single wavelength measurements (Figure 3). Both UV-pH titrations afforded the same p*K*_a_, around 2.43, well within the margin of error.

Next, kynurenine was titrated both with potentiometry and NMR-pH titration (Figure 4). The potentiometric measurement afforded only two p*K*_a_ values. However, the rigorous analysis of the NMR-pH titration revealed that two dissociation steps occur in the acidic pH region. This was evident from the chemical shift response of the aromatic signals: while almost no response is visible in the basic pH region (due to the dissociation step of the primary aliphatic ammonium moiety), there was a considerable response in the acidic region, which can only be due to a proton dissociation occurring on the aromatic ring, i.e., on the aromatic ammonium moiety.

Since the aromatic ^1^H NMR signals showed almost no response in the basic pH region (in fact, only the aromatic signal in meta position to the carbonyl group showed a slight wrong-way shift, which will be elaborated later), it could be reasonably assumed that their response in the acidic pH region is selective for the aromatic amino group. The corollary of this assumption is not so obvious, however: the aliphatic ^1^H NMR signals are relatively close to the aromatic amino group (in terms of covalent distance), especially the methylene group (5 covalent distance), therefore selectivity of aliphatic ^1^H NMR signals in favor of aliphatic ammonium and carboxyl dissociation is dubious. This, however, can be exploited: the titration curve of the methylene group thus affords all three macroconstants, including *K*_a1_, *K*_a2_ in the acidic region (Equation (1)), while the aromatic and αCH ^1^H NMR signals selectively afford the microconstants *K*_aC_^A^ (second step dissociation microconstant of the aromatic ammonium group), *K*_aA_^C^ (second step dissociation microconstant of the carboxyl group) (Equation (2)). This would require the αCH signal to be selective towards the aliphatic ammonium and carboxyl dissociation, which will be demonstrated later. Note that Equation (2) in its current form applies to the titration curves of the aromatic signals; for the αCH signal, Equation (2) is rewritten with *K*_aC_^A^ and *K*_aA_^C^ exchanged. For the curve fitting of Equation (2), the macroconstants are treated as fixed parameters transferred from the fitting of Equation (1).(1)$$\delta_{obs}=\frac{\delta_{L}+\delta_{HL}\cdot K_{a2}\cdot 10^{-pH}+\delta_{H_{2}L}\cdot K_{a1}\cdot K_{a2}\cdot 10^{-2\cdot pH}}{1+K_{a2}\cdot 10^{-pH}+K_{a1}\cdot K_{a2}\cdot 10^{-2\cdot pH}}$$(2)$$\delta_{obs}=\frac{\delta_{L}\cdot \left(1+K_{aC}^{A}\cdot 10^{-pH}\right)+\delta_{HL}\cdot \left(K_{aA}^{C}\cdot 10^{-pH}+K_{a1}\cdot K_{a2}\cdot 10^{-2\cdot pH}\right)}{1+K_{a2}\cdot 10^{-pH}+K_{a1}\cdot K_{a2}\cdot 10^{-2\cdot pH}}$$

The remaining dissociation microconstants (*K*_a_^C^ and *K*_a_^A^) were calculated using Equation (3). Modern theoretical considerations of acid-base microconstant determination can be found in the works of Noszál [37,38] and Borkovec [39,40], while some examples can be found in recent works [36,41].(3)$$pK_{a1}+pK_{a2}=pK_{a}^{A}+pK_{aA}^{C}=pK_{a}^{C}+pK_{aC}^{A}$$

The selectivity of αCH ^1^H NMR signal for the carboxyl group can be demonstrated by comparing the chemical shift response of αCH in the acidic region to that of several other amino acids (see Table 1).

Table 1 shows the known chemical shift response values from αCH and βCH_2_ signals of amino acids (or their derivatives in the literature) in the acidic and basic pH region, corresponding to the responses towards carboxyl and ammonium dissociation, respectively. It can be concluded that the kynurenine methylene (βCH_2_) group is not selective for the carboxyl group as its chemical shift response greatly exceeds that which would be expected from carboxyl dissociation. The selectivity of αCH chemical shift response (lying six covalent bonds away from the primary aromatic ammonium group) can be inferred as acceptable from its chemical shift response in the acidic region (0.471). However, this value is somewhat higher than expected (ca. 0.38). In order to confirm the validity of microconstants afforded by the titration curve of the αCH signal, UV-pH titration of kynurenine was performed (bottom of Figure 4), and an isosbestic point was observed. This indicated that the UV response is selective for the aromatic ammonium dissociation, and the microconstants afforded by this titration are in good agreement with the ones obtained by NMR-pH titration. The UV measurement is, however, burdened by greater uncertainty in the pH determined with glass electrodes, and since these measurements have to be conducted near extreme pH regions, the microconstants determined with NMR are deemed more reliable [47]. No UV-pH titration was therefore considered necessary for 3-hydroxykynurenine since the sufficient selectivity of the αCH signal was demonstrated.

The titration of 3-hydroxykynurenine (Figure 5) was treated the same way as for kynurenine, with the difference that 3-hydroxykynurenine has an overlapping dissociation in the basic pH region as well (the primary aliphatic ammonium dissociation overlaps with the dissociation of the phenol group). The same principle can be applied to the macro- and microconstant determination of kynurenine; only the last two macroconstants were transferred from potentiometric titration data since the latter is more precise than NMR-pH determined values and the chemical shift response of the methylene group towards the phenol dissociation (6 covalent distance) is not expected to be reliable. Otherwise, the microconstants of the basic pH range (*K*_aACN_^O^—phenol group, *K*_aACO_^N^—primary aliphatic ammonium group) were determined by fitting an analog (featuring *K*_a3_ and *K*_a4_) of Equation (2) to the titration data of all ^1^H NMR signals. Note that the fitted p*K*_aACN_^O^ falls very close to the calculated p*K*_aAC_^N^; therefore, to ensure a decreasing sequence of *K*_a_ values in every microscopic protonation pathway, an additional constraint was used during fitting: p*K*_aACN_^O^ be greater than the mean of p*K*_a3_ and p*K*_a4_. The microconstants of the acidic pH range (*K*_aC_^A^—primary aromatic ammonium group, *K*_aA_^C^—carboxyl group) were determined by fitting Equation (2) to the titration data of the aromatic and αCH ^1^H NMR signals, while the first two macroconstants were afforded from the titration data of the methylene signal.

The acid-base microspeciation schemes of kynurenine and 3-hydroxykynurenine (also showing the interactivity parameters determined) are shown in Figure 6, and all determined acid-base constants are compiled in Table 2. In this context, the proton microspeciation scheme depicts the acid dissociation pathways between the various protonated chemical species. In the symbols of site-specific (or microscopic) acid dissociation constants, the superscript denotes the site of dissociation, while superscripts (if any) denote the already dissociated sites. The interactivity parameter (p*ε*) quantifies how much the dissociation of one moiety modifies (usually decreases, unless a very seldom cooperativity phenomenon exists in macromolecules) the dissociation constant of a neighboring site. The p*ε* is calculated by taking the difference of p*K*_aX_^Y^ and p*K*_a_^Y^ microconstants, or equivalently, p*K*_aY_^X^ and p*K*_a_^X^. The relative abundance of kynurenine and 3-hydroxykynurenine microspecies as a function of pH are depicted in Figure 6 next to the respective microspeciation schemes.

To further obtain additional information regarding the microspeciation scheme, the synthesis of methyl ester derivatives was attempted since these model compounds could provide the value of microconstants for cases where the carboxylic acid group is undissociated. However, because of the reactive nature of kynurenine and 3-hydroxykynurenine every attempt at methyl ester formation resulted in decomposition. In the first attempt, the compounds were each dissolved with 1-ethyl-3-(3-dimethylaminopropyl)carbodiimide (EDAC) and triethylamine in dichloromethane, and after fifteen minutes, methanol was added to the mixture, which was stirred overnight at room temperature. After this, the solution was washed with saturated sodium bicarbonate solution and distilled water. The organic phase was dried over sodium sulfate and then removed from the rotary evaporator. In the second attempt, the compounds were each dissolved in methanol, and the solution was cooled to 0 °C. Thionyl chloride was added dropwise, and the mixture was stirred at the same temperature for four hours. Then, the solvent was removed from the rotary evaporator, yielding an oily residue. NMR analysis showed that in both attempts, the initial compounds have decomposed into many products, which were not analyzed further. The measurement of lipophilicity and membrane penetration of kynurenine and 3-hydroxykynurenine was also attempted; however, due to the extremely low predicted log*P* value of these compounds, it was not unexpected that obtaining such experimental data would not be feasible. The in vitro parallel permeability assay (PAMPA) of kynurenine showed no detectable membrane penetration/retention. However, it is important to note that PAMPA only predicts passive transport, while kynurenine is known to be mainly taken up into the brain by carrier proteins [48].

## 4. Discussion

Metabolites of the kynurenine pathway have garnered increasing attention in recent years due to their potential antioxidant and neuroprotective properties [49]. This growing interest underscores the critical need for the development of novel therapeutic strategies targeting neurodegenerative disorders. While the mechanisms of action and biological roles of these metabolites are actively investigated in various experimental models, their fundamental physicochemical characteristics—most notably their acid dissociation constants (p*K*_a_ values)—remain largely uncharacterized through direct experimental methods [50]. Consequently, reliance on inaccurate or estimated p*K*_a_ values may lead to misinterpretations of their redox behavior and, by extension, their functional implications in neurobiology. Therefore, the determination of accurate p*K*_a_ values for kynurenine pathway metabolites may yield insights not only into their acid-base behavior but also their broader redox behavior and reveal notable deviations between computational predictions and experimental data. The current literature on these compounds is limited, and the predicted p*K*_a_ values do not always agree with each other, emphasizing the need for experimental validation of computational models.

One p*K*_a_ of picolinic acid could be determined with accuracy, while the extremely acidic p*K*_a_ value of the carboxyl group was estimated to be closer to 0.6 than 1, as the literature or prediction would suggest. Kynurenic acid exhibited only a single dissociation step in the typical pH range of determination, contrary to what its molecular structure would suggest. This finding indicates that additional protonation or deprotonation sites, while structurally possible, do not contribute significantly under the conditions studied. The behavior of kynurenic acid, in particular, appears to be at odds with its predominant tautomeric form. Based on the simulated NMR spectra (MestReNova) in Figure 3 and the DFT predicted NMR chemical shifts (Jaguar, Schrödinger) in Table 3, the experimental NMR signals of kynurenic acid are somewhat closer to that of the phenolic tautomer, although the number 9 ^1^H vicinal to the carboxyl position (which should be most sensitive to the electrostatic effects of each tautomeric form) lies in between the two simulated values of the tautomers in both methods. Based on these simulations, it can be inferred that kynurenic acid exists in a comparable ratio of its two tautomeric forms without any observed phenol dissociation, which might be caused by the chemical tautomer equilibrium at play. This is supported by experimental data, where only a single p*K*_a_ was determined.

Kynurenine exhibited three dissociation steps, though potentiometric titration was unable to accurately determine the most acidic one. The microscopic p*K*_a_ values derived from NMR titration suggest a moderate interaction between the aromatic ammonium and carboxyl groups, supporting the notion that these functional groups influence each other’s dissociation behavior. This interaction is also evident in 3-hydroxykynurenine, which demonstrated four dissociation steps distributed in two separate pH regions. The microconstants for this compound confirm that the interaction between the aromatic ammonium and carboxyl group is comparable to that observed in kynurenine. However, the interaction between the aliphatic ammonium and the phenol group appears to be mild, indicating that the introduction of the hydroxyl moiety does not substantially alter the acid-base properties of the molecule.

Despite these specific differences, the intrinsic p*K*_a_ values and the sequence of dissociation steps in kynurenine and 3-hydroxykynurenine are in agreement. The presence of the phenolic hydroxyl group in 3-hydroxykynurenine exerts a slight weakening effect on the dissociation of both the aromatic ammonium and carboxylate groups. In contrast, the dissociation of the aliphatic ammonium is slightly enhanced, reflecting a subtle shift in the electronic environment due to the introduction of the hydroxyl group.

The mole fraction abundance curves provide additional insights into the speciation of these metabolites across different pH values. Near the macroscopic p*K*_a_ regions, the experimentally elucidated microspecies contribute significantly to the overall concentration, reinforcing the necessity of detailed microscopic analysis. At physiological pH (7.4), the overwhelming majority of species exist in their zwitterionic form, with the aromatic ring remaining uncharged. This information is particularly relevant in understanding the behavior of these compounds in biological systems, where their charge states influence interactions with proteins, enzymes, and cellular transport mechanisms.

Finally, while the subtle wrong-way chemical shift response observed in kynurenine does not impact the p*K*_a_ determination, it is indicative of an intramolecular hydrogen bond formation. This finding suggests that hydrogen bonding interactions may play a role in stabilizing specific conformations of kynurenine and, analogously, 3-hydroxykynurenin, potentially influencing its biological function and reactivity. Therefore, molecular mechanics and quantum chemistry calculations were used within the Schrödinger quantum chemical platform to gain insight into the conformation and intramolecular interactions of kynurenine and 3-hydroxykynurenine (Figure 7). As expected from the interactivity parameter, there is an intramolecular hydrogen bond between the aromatic amino and carboxylate moieties. The interactivity parameter of these two moieties is higher than expected since the covalent distance would suggest no such interaction. The relative potential energy of this conformer for both kynurenine and 3-hydroxykynurenine are comparable, as are the respective interactivity parameters as well. This intramolecular hydrogen bond is only present in the second most stable conformer; however, the most stable conformer contains two intramolecular hydrogen bonds and for both, the keto group functions as the bond acceptor. Future studies could further explore the biological interactions of these metabolites using advanced spectroscopic and computational methods to refine our understanding of their role in antioxidant defense.

Kynurenine exhibits notable antioxidant and neuroprotective effects, which are at least partially attributable to its capacity for reactive oxygen species (ROS) scavenging and radical trapping [51]. These effects were previously ascribed to its metabolite, kynurenic acid. However, experimental data demonstrates that kynurenine itself can significantly reduce ROS production and lipid peroxidation. It has also been shown to react with hydroxyl radicals and peroxynitrite, yielding kynurenic acid. Additionally, kynurenine is capable of scavenging hydrogen peroxide and superoxide, thereby mitigating oxidative damage. However, kynurenine may exert concentration-dependent dual effects. At elevated concentrations, it induces apoptosis through ROS generation. This phenomenon has been presented as a paradigm shift in the cellular environment-dependent theory of pro-oxidant/antioxidant systems [52]. Kynurenic acid, a downstream metabolite of kynurenine, is primarily recognized for its neuroprotective and antioxidant properties. It scavenges ROS independently of N-methyl-D-aspartate (NMDA) and nicotinic receptor interactions [53]. At concentrations exceeding 100 µM, kynurenic acid can abolish iron(II) sulfate-induced ROS production. Mechanistically, it acts as an endogenous antagonist of ionotropic excitatory amino acid receptors, further contributing to neuroprotection. Despite its beneficial effects, elevated levels of kynurenic acid may exert detrimental impacts on central nervous system function, highlighting the need for tight regulation of its concentration in vivo.

3-Hydroxykynurenine is a complex metabolite with both pro-oxidative and antioxidative activities, depending on its concentration and cellular context. It interacts with xanthine oxidase to produce superoxide radicals, hydrogen peroxide, and hydroxyl radicals at levels sufficient to induce internucleosomal DNA (deoxyribonucleic acid) damage [54]. At low concentrations, 3-hydroxykynurenine elicits cell death via the generation of ROS, while at higher levels, it displays antioxidant capacity by scavenging hydroxyl radicals and peroxynitrite. Picolinic acid possesses non-selective metal ion chelating properties and has demonstrated both neuroprotective and antioxidant effects [55]. Although its efficacy is reported to be lower than that of kynurenic acid, it surpasses that of ascorbic acid. These findings support its potential as a modulatory agent in neuroprotective strategies. It is, therefore, important for the metal ion chelation function that the dominant species of picolinic acid is uncharged under physiological conditions; under slightly acidic media, the protonated charged state of the nitrogen atom would render picolinic acid ineffective. Considering the acidic pH characteristic of inflamed tissues, the antioxidant or neuroprotective effect of picolinic acid could be modulated by decreasing its p*K*_a_ value by carefully designed substituents on the aromatic ring.

The dualistic redox behavior exhibited by kynurenine pathway metabolites highlights the context-dependent nature of their chemical activity. These effects are modulated not only by metabolite concentration but also by the physicochemical characteristics of the cellular environment, including pH, ionic strength, and the presence of endogenous modulators. Consequently, elucidating the specific molecular interactions in which these compounds participate is essential for a comprehensive understanding of their biological functions and for guiding the rational design of novel therapeutic agents. Notably, the pH dependence of the interaction between radical forms of kynurenic acid and tryptophan has been previously demonstrated [23], suggesting that protonation states significantly influence binding affinities and reactivity. Furthermore, the pH-sensitive activity of kynurenine aminotransferase [56] may reflect not only the pH dependence of the enzyme itself but also the altered charge distribution of its substrate near the pH optimum. Such findings imply that subtle shifts in acid-base equilibria can meaningfully impact enzymatic turnover and metabolite reactivity.

The electron density distribution, particularly on the phenolic moieties of these compounds, has also been shown—via density functional theory (DFT) calculations [57]—to influence their capacity to scavenge or generate reactive oxygen species (in the case of 3-hydroxykynurenine). This effect is largely mediated through phenoxyl radical formation and is closely tied to the hydrogen atom transfer potential (formal hydrogen transfer mechanism) and hydrogen bonding capabilities of these functional groups. As such, a detailed understanding of acid-base characteristics is essential to accurately model these redox behaviors under physiological conditions [58]. The conformational structure of 3-hydroxykynurenine, as depicted in Figure 7, reveals that the phenolic hydroxyl group remains unengaged in intramolecular hydrogen bonding. This structural feature allows the phenol moiety to remain accessible for interactions such as hydrogen atom transfer and the formation of phenoxyl radicals. These findings have significant implications for the rational design of 3-hydroxykynurenine, derivatives substituted on the phenolic hydroxyl group, with tailored redox properties and enhanced therapeutic potential. The hydrogen atom transfer mechanism of aromatic amines, particularly those involving the N–H moiety—structurally analogous to that found in kynurenine [59]—has been shown to rely critically on two key factors: the availability of reactive hydrogen atoms and the presence of non-bonded electron pairs on the nitrogen atom (i.e., unprotonated form). These features are essential for effective radical trapping and redox activity. Modulating the acid-base properties of the aromatic amino group, particularly through alteration of its p*K*_a_ or by influencing the strength of hydrogen bonding with neighboring oxo or carboxylate groups, can substantially affect the distribution of protonated and deprotonated species under physiological conditions. Such modulation provides a strategic avenue to fine-tune the antioxidant properties of kynurenine-derived compounds. Given that the protonation of the aromatic amino groups in kynurenine and 3-hydroxykynurenine occurs only under highly acidic conditions—well outside the physiological pH range—direct modulation of their protonation state is unlikely to be a viable strategy for enhancing redox activity in vivo. Instead, efforts to modulate antioxidant potential should focus on manipulating the strength and nature of hydrogen bonding interactions involving adjacent oxo or carboxylate functional groups. Strengthening these hydrogen bonds may stabilize the presence of radicals on the nitrogen atom and the loss of a hydrogen atom, thereby increasing the efficiency of radical scavenging. Such structural modifications could enable the development of novel antioxidant agents with improved specificity and efficacy, grounded in a mechanistic understanding of hydrogen bonding and electron transfer dynamics. Comprehensive characterization of the solution-phase molecular conformations and hydrogen bonding networks of kynurenine metabolites may, therefore, provide critical insights into their reactivity profiles. Further research is warranted, however, to delineate their detailed redox activity mechanism and kinetics under varying chemical environments, which will be crucial for advancing their therapeutic potential.

## 5. Conclusions

This study demonstrates the complex acid-base behavior of kynurenine pathway metabolites and suggests that their interactions and protonation states are vital for understanding their biological roles. Understanding the microspeciation of these metabolites across various pH levels is critical for appreciating their physicochemical behavior. At physiological pH, the predominant zwitterionic form suggests implications for their interactions with biomolecules, influencing processes such as neurotransmission and immune response regulation. Future work could involve deeper studies into intramolecular hydrogen bonding effects observed in the compounds, using advanced methods to further elucidate their biochemical interactions. This thorough investigation provides a robust foundation for further research into the kynurenine pathway’s implications in health and disease.

## Figures and Tables

**Figure 1 antioxidants-14-00589-f001:**
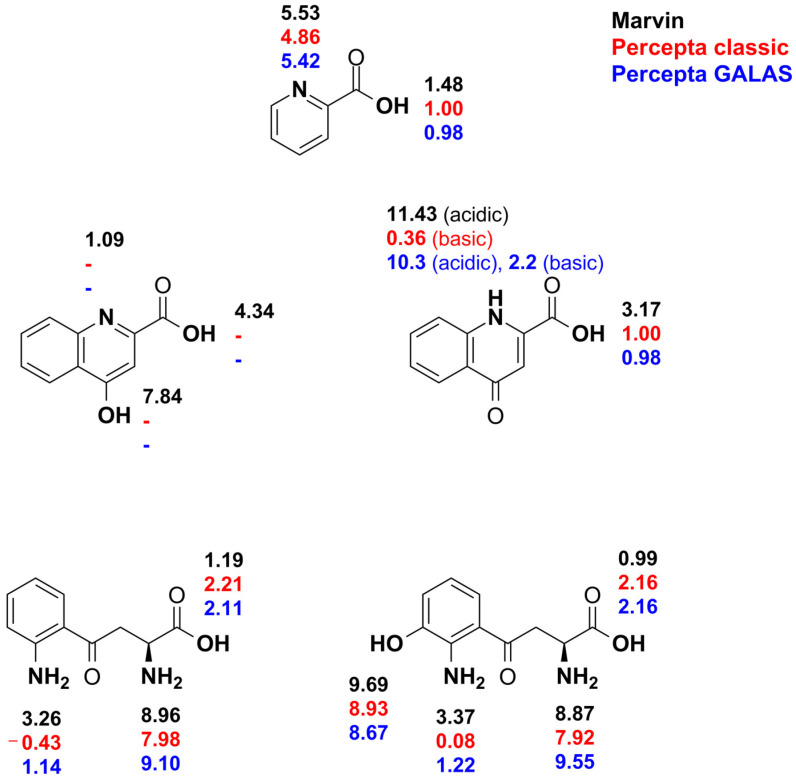
The simulated intrinsic p*K*_a_ values of the studied compounds with the concomitant acid-base moiety are depicted in bold. The color codes show the computational method employed; simulation with Percepta was performed using two methods.

**Figure 2 antioxidants-14-00589-f002:**
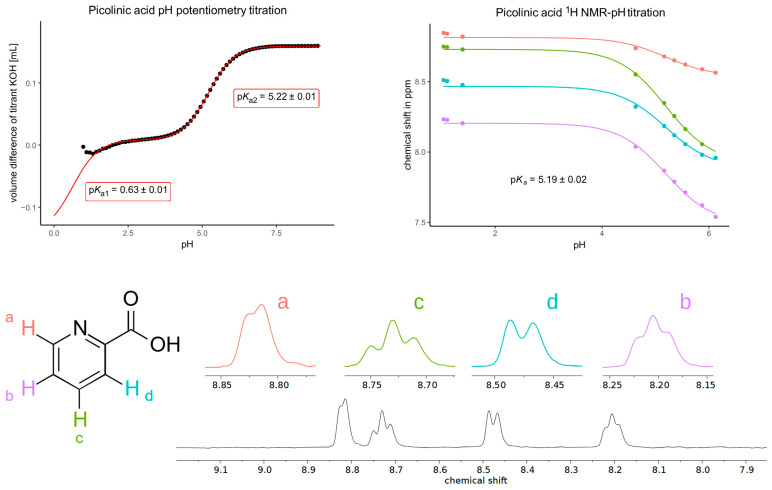
The pH potentiometry titration curve of picolinic acid (**top left**) shows the measured data points in black circles and the fitted curve in red lines. Note that the standard error of p*K*_a2_ is the result of triplicate measurements, while the standard error of p*K*_a1_ is the result of a single nonlinear curve fitting. The NMR-pH titration curve (**top right**) shows the chemical shift response of each nucleus depicted on the structural formula with colors; a characteristic NMR spectrum (recorded at pH 1.39) is also shown (**below**).

**Figure 3 antioxidants-14-00589-f003:**
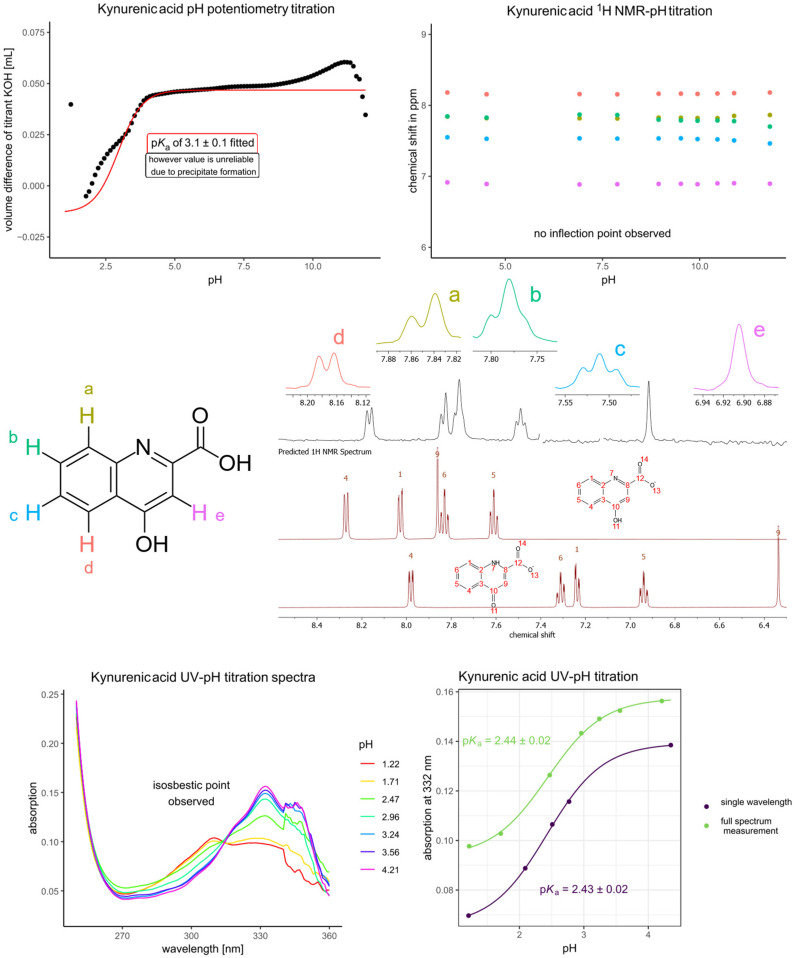
The pH potentiometry titration curve of kynurenic acid (**top left**) shows the measured data points in black circles and the fitted curve in red lines. The NMR-pH titration curve (**top right**) shows the chemical shift response of each nucleus depicted on the structural formula with colors; a characteristic NMR spectrum (recorded at pH 9.52) is also shown (**center**). Each fitted value appearing on a titration curve corresponds to that method of determination (potentiometry or NMR titration). Below the measured NMR spectrum, the simulated spectra (with MestReNova v9.0.1-13254) of the 2 tautomeric forms of kynurenic acid are also shown. Below, the UV spectra of kynurenic acid are shown with varying pH (**bottom left**) together with the UV-pH titration curves afforded (**bottom right**).

**Figure 4 antioxidants-14-00589-f004:**
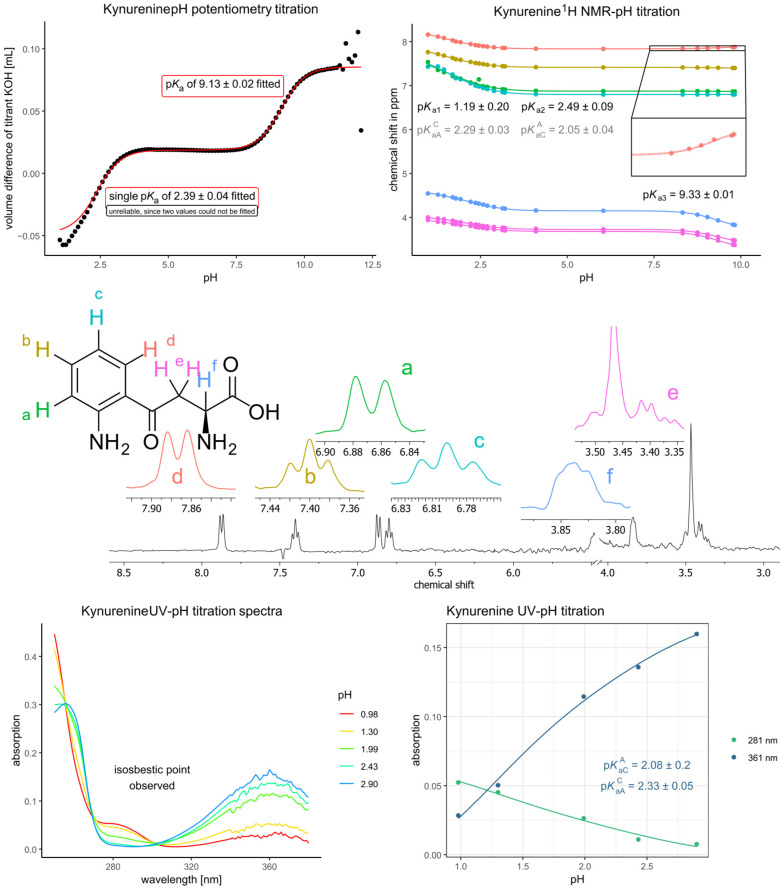
The pH potentiometry titration curve of kynurenine (**top left**) shows the measured data points in black circles and the fitted curve in red lines. The NMR-pH titration curve (**top right**) shows the chemical shift response of each nucleus depicted on the structural formula with colors; a characteristic NMR spectrum (recorded at pH 9.04) is also shown (**center**). Note that the peak of sarcosine (one of the in situ pH indicators) overlaps with the methylene signal of kynurenine at 3.35 ppm. Also, note that the wrong-way shift of the red aromatic signal is barely noticeable but present in the NMR titration curve in the basic pH region. Each fitted value appearing on a titration curve corresponds to that method of determination (potentiometry or NMR titration). Below, the UV spectra of kynurenine are shown with varying pH (**bottom left**) together with the UV-pH titration curves afforded (**bottom right**).

**Figure 5 antioxidants-14-00589-f005:**
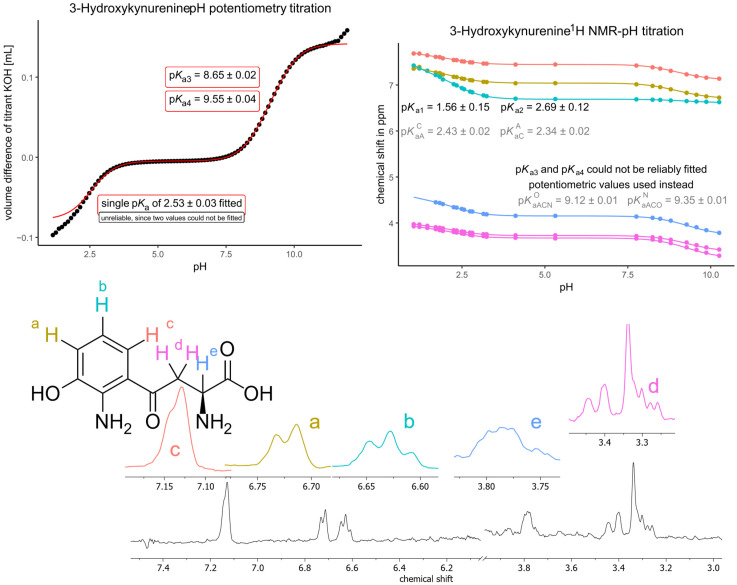
The pH potentiometry titration curve of 3-hydroxykynurenine (**top left**) shows the measured data points in black circles and the fitted curve in red lines. The NMR-pH titration curve (**top right**) shows the chemical shift response of each nucleus depicted on the structural formula with colors; a characteristic NMR spectrum (recorded at pH 10.26) is also shown (**below**). Note that the peak of sarcosine (one of the in situ pH indicators) overlaps with the methylene signal of 3-hydroxykynurenine at 3.30–3.35 ppm. Each fitted value appearing on a titration curve corresponds to that method of determination (potentiometry or NMR titration).

**Figure 6 antioxidants-14-00589-f006:**
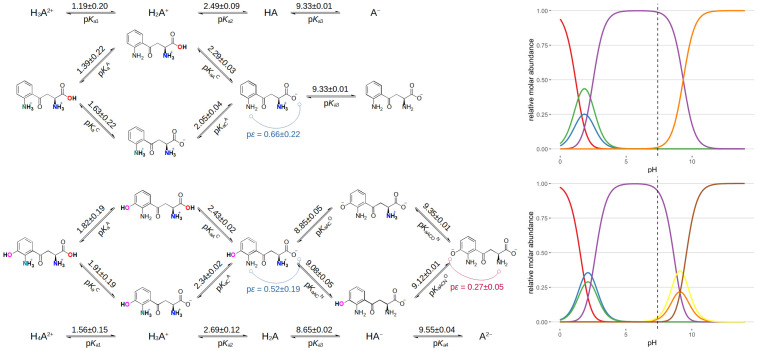
The acid-base microspeciation scheme of kynurenine (**top**) and 3-hydroxykynurenine (**below**) in the direction of proton dissociation (the dissociated proton is not depicted but considered to be inherent to the equilibria). Note that the preferred dissociation path is the upward direction at each junction of the microspeciation scheme. The p*K*_a_ microconstant values are depicted on top of the equilibrium arrows, while their symbol is depicted under their equilibrium arrows. The p*K*_a_ macroconstants are also depicted. The relative abundance of each microspecies is depicted with molar fractions (on the **right**). The colored molar fraction curves correspond to the protonation microscpecies: sequentially, each curve maximum moving from acidic to basic pH corresponds to microspecies moving from the left to the right of the scheme, e.g., the red curve of kynurenine corresponds to the first microspecies that are completely protonated; the green and blue curves correspond to the next two kynurenine microspecies above each other; the purple curve corresponds to the HA species; finally, the orange curve corresponds to the completely deprotonated species.

**Figure 7 antioxidants-14-00589-f007:**
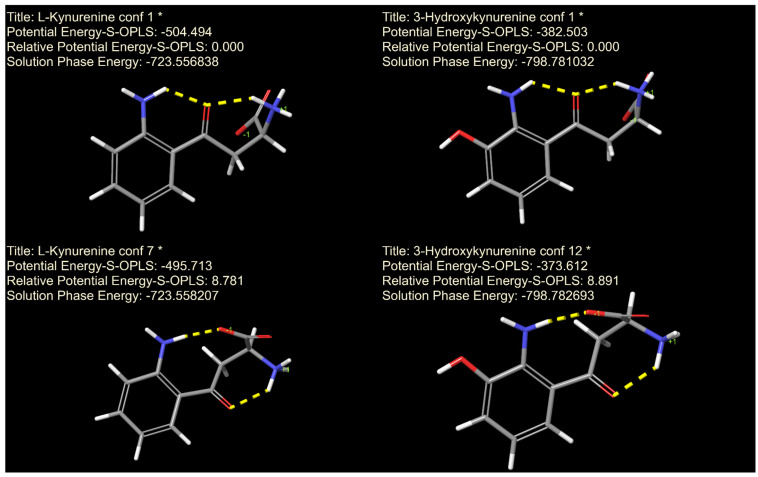
The two most stable characteristic conformers of kynurenine (**left**) and 3-hydroxykynurenine (**right**). Characteristic conformers mean that several of the most stable conformers had the same overall structure. Therefore, only a representative was used for quantum calculations. The intramolecular hydrogen bonds stabilizing the conformers are shown with yellow dashed lines.

**Table 1 antioxidants-14-00589-t001:** The chemical shift response of aliphatic protons in amino acid derivatives: kynurenine (this work) in the basic pH region (first row) corresponds to the aliphatic ammonium dissociation, the acidic pH region (second row in bold) corresponds to the aromatic ammonium and carboxyl dissociations. The other amino acid derivatives only pertain to aliphatic ammonium dissociation (first row) and carboxyl dissociation (second row in bold). ^a^ [42], ^b^ [43], ^c^ [44], ^d^ [45], ^e^ [46].

Kynurenine			Histidine ^a^			Asparagine ^b^		
αCH	βCH_2,a_	βCH_2,b_	αCH	βCH_2,a_	βCH_2,b_	αCH	βCH_2,a_	βCH_2,b_
0.427	0.333	0.405	0.480	-	**-**	0.459	0.302	0.418
**0.471**	**0.313**	**0.290**	**0.407**	-	-	**0.387**	**0.154**	**0.162**
Cysteine ^c^			3-nitrotyrosine ^d^			Serotonin ^e^		
αCH	βCH_2,a_	βCH_2,b_	αCH	βCH_2,a_	βCH_2,b_	αCH	βCH_2,a_	βCH_2,b_
0.532	0.247	0.286	0.474	-	-	0.471	0.318	0.376
**0.380**	**0.090**	**0.107**	**0.376**	**-**	**-**	**0.342**	**0.062**	**0.178**

**Table 2 antioxidants-14-00589-t002:** The compilation of determined p*K*_a_ values is accepted as most reliable in this work, with indications of their method of determination.

Picolinic Acid		Kynurenic Acid	
p*K*_a1_ (by extrapolation of potentiometry)	0.63	p*K*_a1_ (UV)	2.43
p*K*_a2_ (mean of potentiometry and NMR)	5.21		
Kynurenine (NMR)		3-Hydroxykynurenine (NMR, unless stated otherwise)	
p*K*_a1_	1.19	p*K*_a1_	1.56
p*K*_a2_	2.49	p*K*_a2_	2.69
p*K*_a3_	9.33	p*K*_a3_ (potentiometry)	8.65
p*K*_a_^A^	1.39	p*K*_a4_ (potentiometry)	9.55
p*K*_a_^C^	1.63	p*K*_a_^A^	1.82
p*K*_aC_^A^	2.05	p*K*_a_^C^	1.91
p*K*_aA_^C^	2.29	p*K*_aC_^A^	2.34
		p*K*_aA_^C^	2.43
		p*K*_aAC_^O^	8.85
		p*K*_aAC_^N^	9.08
		p*K*_aACN_^O^	9.12
		p*K*_aACO_^N^	9.35

**Table 3 antioxidants-14-00589-t003:** The experimental and simulated ^1^H chemical shift values of kynurenic acid (in its dissociated kynurenate form) pertaining to H_2_O solvent.

Chemical Shifts (ppm)		MestReNova		Jaguar	
^1^H Signal Numbering According to Figure 3	Experimental	Phenol Tautomer	Keto Tautomer	Phenol Tautomer	Keto Tautomer
doublet (4)	8.18	8.27	7.98	7.92	8.13
doublet (1)	7.86	8.03	7.24	7.93	7.44
triplet (6)	7.70	7.83	7.31	7.79	7.63
triplet (4)	7.46	7.61	6.94	7.69	7.34
singlet (9)	6.90	7.86	6.34	7.70	6.35

## Data Availability

The original contributions presented in this study are included in the article/Appendix A. Further inquiries can be directed to the corresponding authors.

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
