# Peer review of "Physicochemical Characterization of Kynurenine Pathway Metabolites"

_antioxidants, 2025, doi:10.3390/antiox14050589_

Round 1
Reviewer 1 Report
This paper presents a detailed analysis of the acid-base equilibrium of metabolites of the kynurenine pathway and experimentally determines pKa values using potentiometry, NMR, and UV. The pKa values determined are outside the physiological pH range, and the pKa values themselves are not directly related to an understanding of human health or disease. The description in this paper contains careless parts that make it difficult to understand the details. In particular, many details regarding the definition of pKa, measurement methods, and analytical techniques are not provided, and the validity of the analytical methods is not demonstrated. However, this paper has academic value as a resource, and the NMR analytical methods employed are of interest. With appropriate revisions, it could be brought to a level acceptable for publication. It has the potential to become a valuable paper for future readers.
1. L29-L30 Abstract "thereby enhancing our understanding of their roles in health and disease and informing potential therapeutic strategies"
This sentence is not directly related to the conclusion of the paper and is an overstatement that may confuse the reader. Either show the relationship between the conclusion of this paper and this sentence with references or cell-based experiments, or delete this sentence.
2. L75-L84 Introduction
In this paragraph, the authors wrote that some of the pKa values determined in this paper have been reported previously, but no specific values are given, only "no reliable data" and "conflicting data". All previously reported experimental values should be listed in Table 2, etc., together with the references and the methods used.
3. L161-L168 Results
In the potentiometric determination of the pKa1 of picolinic acid, the acid side data appear to be excluded from the fitting. If the acid side data were excluded from the fitting under conditions where the transition is barely visible, this would have no physical significance. Furthermore, if the fitting were done to obtain values close to those obtained from simulations, this would be inappropriate as an experimental scientific procedure. Therefore, it should be stated in the main text that the experimental value for pKa1 could not be determined, and the pKa1 value should be removed from Table 2.
4. L218 Results
Equation 1 is duplicated. Delete one of them.
5. L252 Figure 4 and L275 Figure 5
The pKa values shown in the NMR titration panels do not agree with those in the potentiometry titration panels. For example, in Figure 4, the pKa1 value in the potentiometry panel is 2.53, while it is 1.56 in the NMR panel. I recommend using the pKa values given with the method of determination in Figures 2, 3, 4 and Table 2, such as pKa1 = 1.56 ± 0.15 (NMR).
6. L275 Figure 5
The pKa2 and pKa3 values for potentiometry are listed in Table 2 as pKa3 and pKa4, respectively. The same values are used in the NMR titration panel. Check that the values have not been confused. If the values were obtained by a fitting method not described in the paper, describe the method used.
7. L288 Results
The term "microspeciation" is mainly used in biology and is not commonly used in chemistry. Please explain it in the text, citing references.
8. L289 Figure 6
For 3-Hydroxykynurenine, the equilibrium constants between H2A and HA– and between HA– and A2– are written as Ka1 and Ka2, respectively. Are Ka3 and Ka4 typos?
9. L289 Figure 6
In the graph of the fraction on the right side of the figure, the colors of the graph are not explained. This makes it impossible to understand what the graph actually means.
10. L289 Figure 6
The pKaCN value of 3-Hydroxykynurenine (9.15) is smaller than the pKaCNO value (9.05). This creates a local inconsistency. To resolve this inconsistency, either a constraint must be imposed during fitting so that pKaCN is less than pKaCNO, or an equilibrium model that does not assume this intermediate state will be helpful.
11. L296 Table 2
The pKa values of four compounds were determined by different methods; however, it is not always clearly stated which method was used to determine each pKa value or which method was ultimately selected. For example, the average of the potentiometric and NMR values is used for the pKa2 of picolinic acid, but in other cases the NMR value is preferred. To avoid confusion, it is recommended that all values measured by different methods be clearly stated along with the respective methods.
12. L296 Table 2
The reporting of potentiometric pKa values appears to be arbitrary and unconvincing overall. In particular, the pKa1 value for picolinic acid and the pKa3 and pKa4 values for 3-hydroxykynurenine are based on experimental results of low reliability, yet these numerical values are listed in Table 2. On the other hand, the pKa values for kynurenine (pKa1) and 3-Hydroxykynurenine (pKa1) are based on reliable experimental results, but are not included in Table 2. I recommend that all reliable experimental results be clearly stated.
Author Response
Reviewer 1
Major comments
This paper presents a detailed analysis of the acid-base equilibrium of metabolites of the kynurenine pathway and experimentally determines pKa values using potentiometry, NMR, and UV. The pKa values determined are outside the physiological pH range, and the pKa values themselves are not directly related to an understanding of human health or disease. The description in this paper contains careless parts that make it difficult to understand the details. In particular, many details regarding the definition of pKa, measurement methods, and analytical techniques are not provided, and the validity of the analytical methods is not demonstrated. However, this paper has academic value as a resource, and the NMR analytical methods employed are of interest. With appropriate revisions, it could be brought to a level acceptable for publication. It has the potential to become a valuable paper for future readers.
The authors wish to thank the reviewer for the thorough review and helpful suggestions. The comments were incorporated into the revised manuscript.
Detail comments
Comments 1:
- L29-L30 Abstract "thereby enhancing our understanding of their roles in health and disease and informing potential therapeutic strategies"
This sentence is not directly related to the conclusion of the paper and is an overstatement that may confuse the reader. Either show the relationship between the conclusion of this paper and this sentence with references or cell-based experiments, or delete this sentence.
Response 1:
The sentence was deleted.
Comments 2:
- L75-L84 Introduction
In this paragraph, the authors wrote that some of the pKa values determined in this paper have been reported previously, but no specific values are given, only "no reliable data" and "conflicting data". All previously reported experimental values should be listed in Table 2, etc., together with the references and the methods used.
Response 2:
In the literature we found no instances of pKa values for these compounds where the experimental methods were specified. There is therefore no reliable experimental data that can be provided in the table. The only two references [25 and 26] with pKa values reported for picolinic acid mention no experimental methods either, but these values were appended to the manuscript. It should be noted that the second reference contains an incorrect structure for picolinic acid, hence the “conflicting” nature of the data.
Comments 3:
- L161-L168 Results
In the potentiometric determination of the pKa1 of picolinic acid, the acid side data appear to be excluded from the fitting. If the acid side data were excluded from the fitting under conditions where the transition is barely visible, this would have no physical significance. Furthermore, if the fitting were done to obtain values close to those obtained from simulations, this would be inappropriate as an experimental scientific procedure. Therefore, it should be stated in the main text that the experimental value for pKa1 could not be determined, and the pKa1 value should be removed from Table 2.
Response 3:
The pKa1 of picolinic acid often appears in various contexts (see point above) without mention of the source of the data, therefore it seemed imperative a rough experimental estimation be performed. The NMR titration of such an acidic pKa is not possible due to ionic strength constraints. The same is true for potentiometric titration, however in this case, the fact that every dissociation step must be of the same height can be exploited to extrapolate the potentiometric titration curve. The acidic data are therefore not excluded but rather unavailable (censored from the dataset). These points are made clear in the Results section, however a more elaborate explanation was given. No attempt was made to manipulate the extrapolation to adhere to the simulated values, in fact the determined value is roughly 0.4 pKa units below that of simulated values. In Table 2 the reliability of the pKa1 value of picolinic acid is hinted to by listing it as an extrapolated value. The text reads: “it should be noted that the uncertainty of this value is surely greater than the standard error of the regression fit”.
Comments 4:
- L218 Results
Equation 1 is duplicated. Delete one of them.
Response 4:
The error is corrected.
Comments 5:
- L252 Figure 4 and L275 Figure 5
The pKa values shown in the NMR titration panels do not agree with those in the potentiometry titration panels. For example, in Figure 4, the pKa1 value in the potentiometry panel is 2.53, while it is 1.56 in the NMR panel. I recommend using the pKa values given with the method of determination in Figures 2, 3, 4 and Table 2, such as pKa1 = 1.56 ± 0.15 (NMR).
Response 5:
It is true that there is no agreement between some pKa values determined by the two methods. The reason for this is that potentiometry is unable to determine pKa values outside the range of glass electrode reliability, therefore NMR will be more reliable in these cases. It was indeed misleading to mark the unreliable potentiometry fits on the Figures, therefore they were corrected. The fitted values that appear on potentiometry or NMR titration curves, correspond to the concomitant methods, respectively. This was also depicted in the Figure caption.
Comments 6:
- L275 Figure 5
The pKa2 and pKa3 values for potentiometry are listed in Table 2 as pKa3 and pKa4, respectively. The same values are used in the NMR titration panel. Check that the values have not been confused. If the values were obtained by a fitting method not described in the paper, describe the method used.
Response 6:
The labelling of pKa3 and pKa4 for 3-hydroxykynurenine was indeed confused, the pKa2 and pKa3 labelling is incorrect. Figures are corrected and the values in the table are denoted with the methods used for determination.
Comments 7:
- L288 Results
The term "microspeciation" is mainly used in biology and is not commonly used in chemistry. Please explain it in the text, citing references.
Response 7:
The necessary explanation was added to the manuscript.
Comments 8:
- L289 Figure 6
For 3-Hydroxykynurenine, the equilibrium constants between H2A and HA– and between HA– and A2– are written as Ka1 and Ka2, respectively. Are Ka3 and Ka4 typos?
Response 8:
Yes, they are typos and have been corrected in the figure.
Comments 9:
- L289 Figure 6
In the graph of the fraction on the right side of the figure, the colors of the graph are not explained. This makes it impossible to understand what the graph actually means.
Response 9:
Adding to the figure a legend as a key for the color coding would be perhaps too crowded. Therefore, an explanation of the graph was added to the figure caption.
Comments 10:
- L289 Figure 6
The pKaCN value of 3-Hydroxykynurenine (9.15) is smaller than the pKaCNO value (9.05). This creates a local inconsistency. To resolve this inconsistency, either a constraint must be imposed during fitting so that pKaCN is less than pKaCNO, or an equilibrium model that does not assume this intermediate state will be helpful.
Response 10:
Indeed, the pKaACN and pKaACNO fall very close to each other. The fitting of pKaACON and pKaACNO was repeated with adding an additional constraint that pKaACNO be greater than (pKa3 + pKa4)/2. The data, figures and tables were updated.
Comments 11:
- L296 Table 2
The pKa values of four compounds were determined by different methods; however, it is not always clearly stated which method was used to determine each pKa value or which method was ultimately selected. For example, the average of the potentiometric and NMR values is used for the pKa2 of picolinic acid, but in other cases the NMR value is preferred. To avoid confusion, it is recommended that all values measured by different methods be clearly stated along with the respective methods.
Response 11:
Table 2 contains the final values accepted as most reliable together with the methods used indicated.
Comments 12:
- L296 Table 2
The reporting of potentiometric pKa values appears to be arbitrary and unconvincing overall. In particular, the pKa1 value for picolinic acid and the pKa3 and pKa4 values for 3-hydroxykynurenine are based on experimental results of low reliability, yet these numerical values are listed in Table 2. On the other hand, the pKa values for kynurenine (pKa1) and 3-Hydroxykynurenine (pKa1) are based on reliable experimental results, but are not included in Table 2. I recommend that all reliable experimental results be clearly stated.
Response 12:
Table 2 was highlighted to indicate the final values accepted as most reliable in this work.
Reviewer 2 Report
The work is well prepared, presenting a good critical discussion of the results obtained, which may be useful in future investigations on the role of different metabolites.
The only point where a more pertinent problem may arise is the use of a single NMR-pH titration and how the authors managed to perform the entire experiment over the entire pH range without substantial change in sample volume/concentration. Perhaps it would be advisable for the authors to introduce a commentary on this topic.
Authors may want to check whether Equation (1) at line 218 is repeated for a reason or whether this is an error.
All NMR spectra presented do not contain information about the pH value at which they were obtained.
In Figure 5, in the representation of the potentiometric titration there is a text box superimposed on the curve.
Author Response
Reviewer 2
Comments 1:
Major comments
The work is well prepared, presenting a good critical discussion of the results obtained, which may be useful in future investigations on the role of different metabolites.
The only point where a more pertinent problem may arise is the use of a single NMR-pH titration and how the authors managed to perform the entire experiment over the entire pH range without substantial change in sample volume/concentration. Perhaps it would be advisable for the authors to introduce a commentary on this topic.
The authors wish to thank the reviewer for the thorough review and helpful suggestions. The NMR titrations were performed with concentrated titrant solutions (2-5 M HCl or NaOH), therefore dilution of the titrand solution was negligible. Even though some dilution naturally occurs, this does not disrupt the titration, since chemical shifts are not sensitive to concentration and pKa values change only negligibly in this ionic strength range. The necessary comment was added to the Methods section.
Detail comments
Comments 2:
Authors may want to check whether Equation (1) at line 218 is repeated for a reason or whether this is an error.
Response 2:
This is only an error and is corrected.
Comments 3:
All NMR spectra presented do not contain information about the pH value at which they were obtained.
Response 3:
The pH of solutions used for recorded NMR spectra was added to the figure captions.
Comments 4:
In Figure 5, in the representation of the potentiometric titration there is a text box superimposed on the curve.
Response 4:
Figure 5 was updated.
Round 2
Reviewer 1 Report
No
No